# Asymmetric Attributional Word Similarity Measures to Detect the Relations of Textual Generality

**Sebastião Pais** [1,2,3,*,†] and **Gaël Dias** [2,†]

1 Departamento de Informática, Faculdade de Engenharia, Universidade da Beira Interior, 6201-001 Covilhã, Portugal
2 Groupe de Recherche en Informatique, Automatique, Image et Instrumentation (GREYC), National Graduate School of Engineering and Research Center (ENSICAEN), Université de Caen Normandie (UNICAEN), 14000 Caen, France; gael.dias@unicaen.fr
3 NOVA Laboratory for Computer Science and Informatics, Departamento de Informática, Faculdade de Ciências e Tecnologia, Universidade NOVA de Lisboa, 1099-085 Lisboa, Portugal
* Correspondence: sebastiao@di.ubi.pt
† These authors contributed equally to this work.

**Abstract:** In this work, we present a new unsupervised and language-independent methodology to detect the relations of textual generality. For this, we introduce a particular case of Textual Entailment (TE), namely Textual Entailment by Generality (TEG). TE aims to capture primary semantic inference needs across applications in Natural Language Processing (NLP). Since 2005, in the TE Recognition (RTE) task, systems have been asked to automatically judge whether the meaning of a portion of the text, the Text ($T$), entails the meaning of another text, the Hypothesis ($H$). Several novel approaches and improvements in TE technologies demonstrated in RTE Challenges are signaling renewed interest towards a more in-depth and better understanding of the core phenomena involved in TE. In line with this direction, in this work, we focus on a particular case of entailment, entailment by generality, to detect the relations of textual generality. In text, there are different kinds of entailments, yielded from different types of implicative reasoning (lexical, syntactical, common sense based), but here, we focus just on TEG, which can be defined as an entailment from a specific statement towards a relatively more general one. Therefore, we have $T \overset{G}{\to} H$ whenever the premise $T$ entails the hypothesis $H$, this also being more general than the premise. We propose an unsupervised and language-independent method to recognize TEGs, from a pair $\langle T, H \rangle$ having an entailment relation. To this end, we introduce an Informative Asymmetric Measure (IAM) called Simplified Asymmetric InfoSimba (AISs), which we combine with different Asymmetric Association Measures (AAM). In this work, we hypothesize about the existence of a particular mode of TE, namely TEG. Thus, the main contribution of our study is highlighting the importance of this inference mechanism. Consequently, the new annotation data seem to be a valuable resource for the community.

**Keywords:** textual entailment by generality; asymmetric word similarities; asymmetric association measure; informative asymmetric measure

## 1. Introduction

Moving to the realm of Natural Language Processing (NLP), we can analogically perceive inference over information stated in human language. Such inference can be defined as the process of concluding about the truth of a textual statement based on (the truth of) another given piece of text. Textual entailment has captured this language-oriented view on an inference paradigm, initially proposed by [1] and subsequently established through the series of benchmarks known as the *PASCAL Recognising Textual Entailment (RTE) Challenges*.



While capturing a generic notion of inference over texts, the introduction of entailment recognition as a computational task was mainly motivated by its overarching potential for NLP applications. For example, consider a Question-Answer (QA) scenario, addressing the question *"Who painted 'The Scream'?"*. In order to provide the answer *"Edvard Munch"*, based on the text snippet *"Norway's most famous painting, 'The Scream' by Edvard Munch, …"*, the QA system needs to validate that the hypothesized answer statement *"Edvard Munch painted 'The Scream'."* is indeed entailed (inferred) by the given text.

Entailment is widely used in many aspects of human life. Assume that someone is seeking something, and he or she searches for the answer from books, friends, or the Web. In most cases, the information gathered or retrieved is not the exact answer, although the (information) seeker may have one idea in his or her mind. Instead, the consequences of the original goal may be detected, so entailment plays a role and confirms or denies the original information being sought [2].

Textual entailment, being a semantic relation, is also one such fundamental keystone with relevant implications and utilities in different NLP areas and is mostly required to overcome some challenging tasks. For instance, it has been applied in text summarization [3,4], machine translation [5], and more recently, the concept of chatbots [6,7].

Given the multiple applications that textual entailment can have, we understand that there are several types of implications, where each type of implication stems from or suits a specific task. Clear evidence of this is the diversity of methodologies and results presented in the RTE challenges.

In this paper, we introduce the new concept of textual **Entailment by Generality**. This new paradigm can be defined as the relation that holds between a specific statement that implies a more general one, for example, *strawberry* (specific) implies *fruit* (general), because *strawberry* is a *fruit*, but the *fruit* does not necessarily imply *strawberry*, because, *fruit* can be *strawberry* but can also be *orange*, *banana*, or other *fruit*.

Also, in this paper, we present our methodology—unsupervised, language-independent and threshold-free—for learning to identify entailment by generality between two sentences.

### 1.1. Defining Textual Entailment

We argue that there are several types of entailment, for example, in the study done in [8], they present three types of entailment that can be defined as follows:

1. *Semantic Subsumption—T* and *H* express the same fact, but the situation described in *T* is more specific than the situation in *H*. The specificity of *T* is expressed through one or more semantic operations. For example, in the sentential pair:

   - *H*: The cat eats the mouse.
   - *T*: The cat devours the mouse.

   *T* is more specific than *H*, as *eating* is a semantic generalization of *devouring*.

2. *Syntactic Subsumption – T* and *H* express the same fact, but the situation described in *T* is more specific than the situation in *H*. The specificity of *T* is expressed through one or more syntactic operations. For example, in the pair:

   - *H*: The cat eats the mouse.
   - *T*: The cat eats the mouse in the garden.

   *T* contains a specializing prepositional phrase.

3.     *Direct Implication—H* expresses a fact that is implied by a fact in *T*. For example:

> - *H*: The cat killed the mouse.
> - *T*: The cat devours the mouse.
>
> *H* is implied by *T*, as it is supposed that killed is a precondition for devouring. In [1] syntactic subsumption roughly corresponds to the restrictive extension rule, while the direct implication and semantic subsumption to the axiom rule.

In [8], despite the two types of subsumption entailment, direct implication underlies deeper semantic and discourse analysis. In most cases, as implication concerns two distinct facts in *T* and *H*, and as facts are usually expressed through verbs, it follows that the implication phenomenon is strictly tied to the relationship among the *T* and *H* verbs. In particular, it is interesting to notice the temporal relation between *T* and *H* verbs, as described in [9]. The two verbs are said to be in temporal inclusion when the action of one verb is temporally included in the action of the other (e.g., snore → sleep). Backward-presupposition stands when the *H* verb happens before the *T* verb (win entails play). In causation, a stative verb in *H* necessarily follows a verb of change in *T* (e.g., give → have). In this case, the temporal relation is thus inverted concerning backward-presupposition. Such considerations leave space to the application of temporal and verb analysis techniques both in the acquisition and recognition tasks.

Ultimately, we want to regard entailment by generality as a relation between utterances (that is, sentences in context), where the context is relevant to understand the meaning. Then, considering the study in [8], we understand that the relation entailment by generality can be compared with one of three relations:

- *Semantic Subsumption*;
- *Syntactic Subsumption*;
- Or a combination of both—*Semantic Subsumption + Syntactic Subsumption*;

Finally, in [10], the authors present a review of the logical and philosophical issues involved in providing an adequate definition of textual entailment. They show that many subjective definitions of textual entailment is refuted by counterexamples, including the most widely cited definition of [11]. They then articulate and defend the following revised definition: T textually entails, if a human reading T would be justified in inferring the proposition expressed by H from the proposition expressed by T. They also show that textual entailment is context-sensitive, nontransitive, and nonmonotonic.

Thus, they have argued that, while the inferential approach to defining textual has clear advantages over logical approaches, the standard definition proffered by Dagan and collaborators needs refinement. They have articulated and defended a refined definition, still in the spirit of Dagan et al.'s, according to which a text *T textually entails a hypothesis H relative to a group of end-users G just in case, typically, a member of G reading T would be justified in inferring the proposition expressed by H from the proposition expressed by T*, more formally:

- *T textually entails H relative to group G, if a member of G reading T would be justified in inferring the proposition expressed by H from the proposition expressed by T.*

They have shown how their definition improves upon existing definitions, and they have defended it against a range of objections. Finally, they argued that textual entailment is context-sensitive, nontransitive, and nonmonotonic. This clarification of the notion of textual entailment may be considered, more generally, as an exercise in the conceptual foundations of information science.

For us, in the most common definition, *Entailment by Generality* can be defined as the entailment from a specific sentence towards a more general sentence [12,13].

### 1.2. Textual Entailment by Generality

As the contribution of this work, we introduce the TEG paradigm, which can be defined as the entailment from a specific sentence towards a more general one. For instance, given a pair of sentences $\langle S_1, S_2 \rangle$, as shown in the example below (One example from the RTE-1 corpus), it is naturally evident that $S_1$ entails/implies $S_2$ and the latter is a more general sentence. Therefore, we have TEG from $S_1$ to $S_2$, represented as $S_1 \xrightarrow{G} S_2$.

$S_1$:　*Mexico City has a terrible pollution problem because the mountains around the city act as walls and block in dust and smog.*

$S_2$:　*Poor air circulation out of the mountain-walled Mexico City aggravates pollution.*

To understand how TE by Generality can be modeled, we propose a new paradigm based on a new *Informative Asymmetric Measure* (IAM), called the *Asymmetric InfoSimba Similarity* (AIS) measure. Instead of relying on the exact matches of words between texts, we propose that one sentence entails the other one by generality if two constraints hold: (a) if and only if both sentences share many related words and (b) if most of the words of a given sentence are more general than the words of the other one. As far as we know, we are the first to propose an unsupervised, language-independent, threshold-free methodology in the context of entailment by generality (TEG).

### 1.3. Asymmetric Association

Most of the existing measures to evaluate the degree of similarity between words are symmetric [14,15]. According to [16], similar asymmetry effects are observed in human intuitions about associated words, which can be measured, for example, with free association tasks. For instance, consider the pair $\langle mango, fruit \rangle$. When hearing the word *mango*, the *fruit* is one of the first associations that come to mind. However, when hearing *fruit*, more familiar fruits like *apple* are more likely to be the first associations rather than a less frequent fruit like *mango*. We call *fruit* → *mango* a forward association of the pair $\langle fruit, mango \rangle$ and a backward association of $\langle mango, fruit \rangle$. In this case, the forward association of $\langle fruit, mango \rangle$ is weak, whereas its backward association is more robust. There are several reasons why human associations can be asymmetric, in our case, to detect relations of textual generality, the reason for asymmetry is the degree of generality of the term. There is a tendency for a strong forward association from a specific term like *adenocarcinoma* to the more general term cancer, whereas the association from cancer to *adenocarcinoma* is weak.

In order to keep language-independent and to some extent, propose unsupervised methodologies, different works propose to use asymmetric association measures. Some have been introduced in the domain of taxonomy construction [17], others in cognitive psycholinguistics [16] and word order discovery [18].

In [17] we have one of the first studies to propose the use of *conditional probability*, for taxonomy construction. They assume that a term $t_2$ subsumes a term $t_1$ if the documents in which $t_1$ occurs are a subset of the documents in which $t_2$ occurs constrained by $P(t_2|t_1) \geq 0.8$. By gathering all subsumption relations, they build the semantic structure of any domain, which corresponds to a directed acyclic graph. In [19], the subsumption relation is relieved to the following expression $P(t_2|t_1) \geq P(t_1|t_2)$ and $P(t_2|t_1) > t$ where $t$ is a given threshold and all term pairs found to have a subsumption relationship are passed through a transitivity module, which removes extraneous subsumption relationships in the way that transitivity is preferred over direct pathways, thus leading to a non-triangular directed acyclic graph.

In [16], the plain *conditional probability* and the *ranking measure*, based on the Pearson's $\chi^2$ test, were used as a model for directed psychological association in the human mind. In particular, $R(t_2\|t_1)$ returns the rank of $t_2$ in the association list of $t_1$ given by the order obtained with the Pearson's $\chi^2$ test for all the words co-occurring with $t_1$. So, when comparing $R(t_2\|t_1)$ and $R(t_1\|t_2)$, the smaller rank indicates the strongest association.

In the specific domain of word order discovery, [18] proposed a methodology, combining directed graphs with the TextRank algorithm [20], to automatically induce a general-specific word order for a given vocabulary based on Web corpora frequency counts.

## 2. Related Work

### 2.1. Recognizing Textual Entailment

In 2005, the RTE Challenge was launched by [11], defining TE as a task for automatic systems. Given two texts $T$ and $H$, the task consists in deciding whether the meaning of $H$ can be inferred from the meaning of $T$. The following example shows a $\langle T, H \rangle$ pair for which the entailment relation holds:

- $T$: In the end, defeated, Antony committed suicide and so did Cleopatra, according to legend, by putting an asp to her breast.
- $H$: Cleopatra committed suicide.

At present, *textual entailment* (TE) is considered an interesting and challenging topic within the NLP community, due to its many potential applications. The PASCAL Network promoted a generic evaluation framework covering semantic-oriented inferences for several NLP applications, which led to launching the RTE Challenge. Many research areas such as *Information Extraction*, *Question Answer*, *Information Retrieval*, *Text Summarization* and *Machine Translation* have to cope with different kinds of inference mechanisms, closely related to the entailment notion. In this direction, some works attempted to apply textual entailment to various NLP tasks in order to benefit from a semantic inference framework, and to improve their performances [21] potentially.

Appendix A and Table A1 presents the main approaches used in the first five RTE Challenges, the results obtained by the RTE systems demonstrate general improvement with time, with overall accuracy levels ranging from 50% to 65% on RTE-1 (17 submissions), from 53% to 75% on RTE-2 (23 submissions), from 49% to 80% on RTE-3 (26 submissions), from 45% to 74% on RTE-4 (26 submissions, three-way task) and from 43% to 75% on RTE-5 (20 submissions, three-way task). Additional in Table 1 shows the average of the top five results in RTE Challenges. Conventional approaches used by the submitted systems include ML, logical inference, cross-pair similarity measures between $T$ and $H$ and word alignment.

**Table 1.** Average of the top five results.

| Challenge | Accuracy Average |
|-----------|------------------|
| **RTE-1** | 0.581 |
| **RTE-2** | 0.675 |
| **RTE-3** | 0.711 |
| **RTE-4** | 0.688 |
| **RTE-5** | 0.679 |

### 2.2. Unsupervised Language-Independent Methodologies for RTE

Different approaches have been proposed to recognize Textual Entailment: from unsupervised language-independent methodologies [22–24] to in-deep linguistic analysis. We will significantly detail the unsupervised language-independent approaches, to which our work can be directly compared, at least to a certain extent. One of the most simple proposals [23] explores the *BLEU algorithm* [25]. First, for several values of $n$ (typically from 1 to 4), they calculate the percentage of $n$-grams from the text $T$, which appear in the hypothesis $H$. The frequency of each $n$-gram is limited to the maximum frequency with which it appears in any text $T$. Then, they combine the marks obtained for each value of $n$, as a weighted linear average and finally apply a brevity factor to penalize short texts $T$. The output of BLEU is then taken as the confidence score. Finally, they perform an optimization procedure to choose the best threshold according to the percentage of success of correctly recognized entailments.

The value obtained was 0.157. Thus, if the BLEU output is higher than 0.157, the entailment is marked as true, otherwise as false. This procedure achieves and accuracy of 0.495 in recognizing TE.

In [24], the entailment data is treated as an aligned translation corpus. In particular, they use the *GIZA++* toolkit [26] to induce alignment models. However, the alignment scores alone were next to useless on the RTE-1 development data, predicting entailment correctly only slightly above chance. As a consequence, they introduced a combination of metrics intended to measure translation quality. Finally, they combined all the alignment information and string metrics with the classical K-NN classifier to choose, for each test pair, the dominant truth value among the five nearest neighbors in the development set. This method achieves a 0.586 accuracy.

The most interesting work is certainly the one described in [22], in which the authors propose a general probabilistic setting that formalizes the notion of TE. Here, they focus on identifying when the lexical elements of a textual hypothesis $H$ are inferred from a given text $T$. The probability of lexical entailment is derived from Equation (1), where $hits(.,.)$ is a function that returns the number of documents, which contain its arguments.

$$P(H|T) = \prod_{u \in H} max_{v \in T} \frac{hits(u,v)}{hits(v)} \tag{1}$$

The text and hypothesis of all pairs in the development and test sets were tokenized and stop words were removed to tune a decision threshold, $\lambda$ empirically. Thus, for a pair $\langle T, H \rangle$, they tagged an example as true (i.e., entailment holds) if $P(H|T) > \lambda$, and as false otherwise. The threshold was empirically set to 0.005. With this method, the accuracy of 0.586 is achieved. The best results from these three approaches are obtained by [22], who introduces the notion of asymmetry within their model without clearly mentioning it. The underlying idea is based on the fact that for each word in $H$, the best asymmetrically co-occurring word in $T$ is chosen to evaluate $P(H|T)$. Although all three approaches show interesting properties, they all depend on tuned thresholds, which can not reliably be reproduced and need to be changed for each new application. Moreover, they need training data, which may not be available. Our idea aims at generalizing the hypothesis made by [22]. Indeed, their methodology is only based on one pair $(u,v), \forall u$ and does not take into account the fact that many pairs, i.e., $(u,v), \exists v \forall u$ may help the decision process. Moreover, they do not propose a solution for the case where the ratio $\frac{hits(u,v)}{hits(v)}$ is null. Finally, we propose to avoid the definition of a "hard" threshold and study asymmetry exhaustively in language, i.e., not just by the conditional probability as done in [22]. For that purpose, we propose a new Informative Asymmetric Measure called the Asymmetric InfoSimba Similarity combined with different Association Measures.

## 3. Asymmetric Similarity

Asymmetric Association Measures (AAM) is inspired by the fact that within the human mind, the association between two words or concepts is not always symmetric. For pairs like fruit and apple, one would agree that there is a strong mutual association between the two. When thinking of fruit, it is not very far-fetched to think of an apple as well and vice versa. There are other pairs, however, that do not exhibit this kind of strong association in both directions. Think of the pair fruit and mango, for example. Mango is probably not the first thing that comes to one's mind when hearing the word fruit. On the other hand, mango is strongly associated with the concept of fruit. An example from [16] reads: *"there is a tendency for a strong forward association from a specific term like adenocarcinoma to the more general term cancer, whereas the association from cancer to adenocarcinoma is weak"*. According to [16], this idea bears some resemblance to the prototype theory [27], where objects are regarded as members of different categories. Some members of the same category are more central than others making them more prototypical of the category they belong to. For instance, *cancer* would be more central than *adenocarcinoma*. However, we deeply believe that the main background for the direction of the association lies in the notion of specific and general terms. Indeed, it is clear that there exists a tendency for a strong forward association from a specific term to the more general term, but the

backward association is weaker. Within this scope, several recent works have proposed the use of asymmetric similarity measures. We believe that this idea has the potential to bring about significant improvements in the acquisition of word semantic relations.

The idea of an asymmetric measure is inspired by the fact that within the human mind, the association between two words or concepts is not always symmetric. For example, as stated in [16], "*there is a tendency for a strong forward association from a specific term like adenocarcinoma to the more general term cancer, whereas the association from cancer to adenocarcinoma is weak*". For instance, *cancer* would be more central than *adenocarcinoma*. Within this scope, seldom new researches have been emerging over the past few years, which propose the use of asymmetric similarity measures, which we believe can lead to significant improvements in the acquisition of word semantic relations, as shown in [28].

We present the eight asymmetric association measures used in this work that will be evaluated in the context of asymmetry between sentences: the Added Value (Equation (3)), the Braun-Blanket (Equation (4)), the Certainty Factor (Equation (5)), the Conviction (Equation (6)), the Gini Index (Equation (7)), the J-measure (Equation (8)), the Laplace (Equation (9)), and the Conditional Probability (Equation (2)).

In the context of sentences, there are several ways to compute the similarity between two sentences. Most similarity measures determine the distance between two vectors associated with two sentences (i.e., the vector space model). However, when applying the standard similarity measures between two sentences, only the identical indexes of the row vector $X_i$ and $X_j$ are taken into account, which may lead to miscalculated similarities. To deal with this problem, different methodologies have been proposed, but the most promising one is undoubtedly the one proposed by [29], the InfoSimba informative similarity measure, expressed in Equation (11).

Although there are many asymmetric similarity measures between words, there does not exist any attributional similarity measure capable of assessing whether a sentence is more specific/general than another one. To overcome this issue, we introduce the asymmetric InfoSimba similarity measure (*AIS*), in which the underlying idea is to say that a sentence $T$ is semantically related to sentence $H$ and $H$ is more general than $T$ (i.e., $T \overset{G}{\rightarrow} H$) if $H$ and $T$ share as many relevant related words as possible between contexts and each context word of $H$ is likely to be more general than most of the context words of $T$. The *AIS* is defined in Equation (14).

As the computation of the *AIS* may be hard due to orders of complexity, we also define its simplified version $AISs(.\|.)$ in Equation 15, which we will correctly use in our experiments.

As a consequence, our hypothesis is an entailment by generality ($T \overset{G}{\rightarrow} H$) will hold if and only if $AISs(T\|H) < AISs(H\|T)$. Otherwise, the entailment will not hold. This way, contrary to existing methodologies, we do not need to define or tune thresholds. Indeed, due to its asymmetric definition, the asymmetric InfoSimba similarity measure allows comparing both sides of entailments.

## 4. Detect Relations of Textual Generality

### 4.1. Asymmetric Association Measures

In order to stay within the domain of language-independent and unsupervised methodologies, several asymmetric association measures have been proposed [14,15] and applied to the problems of taxonomy construction [17,30], cognitive psycholinguistics [16] and general-specific word order induction [18].

Reference [17] is undoubtedly one of the first studies to propose the use of the conditional probability, Equation (2), for taxonomy construction.

$$P(x|y) = \frac{P(x,y)}{P(y)}. \tag{2}$$

They assume that a term $t_2$ subsumes a term $t_1$ if the documents in which $t_1$ occurs are a subset of the documents in which $t_2$ occurs constrained by $P(t_2|t_1) \geq 0.8$ and $P(t_1|t_2) < 0.1$. By gathering

all subsumption relations, they build the semantic structure of any domain, which corresponds to a directed acyclic graph. In [19], the subsumption relation is relieved to the following expression $P(t_2|t_1) \geq P(t_1|t_2)$ and $P(t_2|t_1) > \tau$ where $\tau$ is a given threshold and all term pairs found to have a subsumption relationship are passed through a transitivity module, which removes extraneous subsumption relationships in the way that transitivity is preferred over direct pathways, thus leading to a non-triangular directed acyclic graph.

In the specific domain of word order discovery, the authors of [18] propose a methodology based on directed graphs and the TextRank algorithm [20] to automatically induce a general-specific word order for a given vocabulary based on Web corpora frequency counts. A directed graph is obtained by keeping the edge, which corresponds to the maximum value of the asymmetric association measure between two words. Then, the TextRank is applied and produces an ordered list of nouns, on a continuous scale, from the most general to the most specific. Eight of the AAM used in that work will be evaluated in the context of asymmetric similarity between sentences: the Added Value (Equation (3)), the Braun-Blanket (Equation (4)), the Certainty Factor (Equation (5)), the Conviction (Equation (6)), the Gini Index (Equation (7)), the J-measure (Equation (8)), the Laplace (Equation (9)) and the Conditional Probability (Equation (2)).

$$AV(x\|y) = P(x|y) - P(x). \tag{3}$$

$$BB(x\|y) = \frac{f(x,y)}{f(x,y) + f(\bar{x},y)}. \tag{4}$$

$$CF(x\|y) = \frac{P(x|y) - P(x)}{1 - P(x)}. \tag{5}$$

$$CO(x\|y) = \frac{P(x) \times P(\bar{y})}{P(x,\bar{y})}. \tag{6}$$

$$GI(x\|y) = \quad P(y) \times (P(x|y)^2 + P(\bar{x}|y)^2) - P(x)^2 \\ P(\bar{y}) \times (P(x|\bar{y})^2 + P(\bar{x}|\bar{y})^2) - P(\bar{x})^2. \tag{7}$$

$$JM(x\|y) = P(x,y) \times \log \frac{P(x|y)}{P(x)} + P(\bar{x},y) \times \log \frac{P(\bar{x}|y)}{P(\bar{x})}. \tag{8}$$

$$LP(x\|y) = \frac{N \times P(x,y) + 1}{N \times P(y) + 2}. \tag{9}$$

### 4.2. Asymmetric Attributional Word Similarities

In [31], it was noted that it is unjustified from a linguistic point of view to assume that all the dimensions of a vector space model to be orthogonal to each other. Since each dimension typically corresponds to a context word, this is equivalent to the assumption that every two words denote disparate meanings. Such a vector space model fails to account adequately for similar contexts in meaning or synonymous contexts.

The InfoSimba (IS) aims to measure the correlations between all the pairs of words in two-word context vectors instead of just relying on their exact match as with the cosine similarity measure (Equation (10)). Further, IS guarantees to catch similarity between pairs of words, even when they do not share contexts, due to data sparseness, for example. Nevertheless, they have similar contexts. It is defined in Equation (11) where $S(.,.)$ is any symmetric similarity measure, and each $W_{ij}$ corresponds to the attribute word at the $j^{th}$ position in the vector $X_i$, $p$ is the length of the vector $X_i$. Equation (12) is the simplified version $Ss(.,.)$.

$$cos(X_i, X_j) = \frac{\sum_{k=1}^{p} X_{ik} \times X_{jk}}{\sqrt{\sum_{k=1}^{p} X_{ik}^2} \times \sqrt{\sum_{k=1}^{p} X_{jk}^2}}. \tag{10}$$

$$IS(X_i, X_j) = \frac{\sum_{k=1}^{p} \sum_{l=1}^{p} X_{ik} \times X_{jl} \times S(W_{ik}, W_{jl})}{\begin{pmatrix} \sum_{k=1}^{p} \sum_{l=1}^{p} X_{ik} \times X_{il} \times S(W_{ik}, W_{il}) + \\ \sum_{k=1}^{p} \sum_{l=1}^{p} X_{jk} \times X_{jk} \times S(W_{jk}, W_{jl}) - \\ \sum_{k=1}^{p} \sum_{l=1}^{p} X_{ik} \times X_{jl} \times S(W_{ik}, W_{jl}) \end{pmatrix}}. \tag{11}$$

$$ISs(X_i, X_j) = \frac{1}{p^2} \sum_{k=1}^{p} \sum_{l=1}^{p} X_{ik} \times X_{jl} \times S(W_{ik}, W_{jl}) \tag{12}$$

In the context of asymmetric attributional word similarities research [32,33] the directions of co-occurrences are noted and exploited, but there does not exist an in-depth study either a theoretical account of this phenomenon. The efforts are directed towards developing asymmetric distributional similarity measures such as the Kullback–Leibler divergence [34] defined in Equation (13) where $A = \{\langle z, r \rangle | \exists(x, z, r) \wedge \langle z, r \rangle | \exists(y, z, r)\}$, which has been regularly set apart from the Jensen–Shannon divergence [35], its symmetric counterpart. We can also point at the cross-entropy described in [14].

$$KL(x\|y) = \sum_{\langle z,r \rangle \in A} \log P(z|x) \times \frac{\log P(z|x)}{\log P(z|y)}. \tag{13}$$

Although there are many asymmetric similarity measures, they evidence problems that may reduce their utility. On the one hand, asymmetric association measures can only evaluate the generality/specificity relation between words that are known to be in a semantic relation such as in [17,18]. Indeed, they generally capture the direction of the association between two words based on document contexts and only take into account loose semantic proximity between words. For example, it is highly probable to find that *Apple* is more general than the *iPad*, which can not be assimilated to a hypernymy/hyponymy or meronymy/holonymy relation. On the other hand, asymmetric attributional word similarities only take into account common contexts to assess the degree of asymmetric relatedness between two words. To leverage these issues, we define AIS measure, which the underlying idea is to say that one-word $x$ is semantically related to a word $y$ and $x$ is more general than $y$ if $x$ and $y$ share as many relevant related words as possible and each context word of $x$ is likely to be more general than most of the context words of $y$. The AIS is defined in Equation (14), where $AS(.\|.)$ is an asymmetric similarity measure, likewise for the IS in Equation (11) where $S(.,.)$ stands for any symmetric similarity measure, where each $W_{ik}$ and $W_{jl}$ corresponds to the attribute word at the $k^{th}$ and $l^{th}$ position in the vectors $X_i$ and $X_j$ respectively, $p$ is the length of the vector $X_i$ and $X_j$ (the lengths of the vectors can be different, so, $p$ can be different from the accord with a vector). We also define its simplified version $AISs(.\|.)$ in (15).

$$AIS(X_i\|X_j) = \frac{\sum_{k=1}^{p} \sum_{l=1}^{p} X_{ik} \times X_{jl} \times AS(W_{ik}\|W_{jl})}{\begin{pmatrix} \sum_{k=1}^{p} \sum_{l=1}^{p} X_{ik} \times X_{il} \times AS(W_{ik}\|W_{il}) + \\ \sum_{k=1}^{p} \sum_{l=1}^{p} X_{jk} \times X_{jk} \times AS(W_{jk}\|W_{jl}) - \\ \sum_{k=1}^{p} \sum_{l=1}^{p} X_{ik} \times X_{jl} \times AS(W_{ik}\|W_{jl}) \end{pmatrix}}. \tag{14}$$

$$AISs(X_i\|X_j) = \frac{1}{p^2} \sum_{k=1}^{p} \sum_{l=1}^{p} X_{ik} \times X_{jl} \times AS(W_{ik}\|W_{jl}),$$
$$\text{provided that: } W_{ik} \neq W_{jl}. \tag{15}$$

## 5. TEG Corpus

Large scale annotation projects such as TreeBank [36], PropBank [37], TimeBank [38], FrameNet [39], SemCor [40], and others play an important role in NLP research, encouraging the development of new ideas, tasks, and algorithms. The construction of these datasets, however, is costly

in both annotator-hours and financial cost. Since the performance of many NLP tasks is limited by the amount and quality of data available to them [41], one promising alternative for some tasks is the collection of non-expert annotations. The availability and the increasing popularity of crowdsourcing services have been considered as an interesting opportunity to meet the needs above and design criteria.

Crowdsourcing services have been recently used with success for a variety of NLP applications [42]. Although Amazon Mechanical Turk (MTurk) is directly accessible only to US citizens, the *Figure Eight Platform* service (https://www.figure-eight.com/ [Last access: 28 June 2019]) provides a crowdsourcing interface to MTurk for non-US citizens.

The main idea in using crowdsourcing to create NLP resources is that the acquisition and annotation of large datasets, needed to train and evaluate NLP tools and approaches, can be carried out cost-effectively by defining simple Human Intelligence Tasks (HITs) routed to a crowd of non-expert workers, called "*Turkers*", hired through online marketplaces.

*Building Methodology—Quantitative Analysis*

Our approach builds on a pipeline of HITs routed to the MTurk workforce through the *Figure Eight Platform* interface. The objective is to collect $\langle T, H \rangle$ pairs where entailment by generality holds.

Our building methodology has several stages. First, we select the positive pairs of TE from the first five RTE challenges. These pairs are then submitted to *Figure Eight Platform* through a job that we have built online, to be evaluated by *"Turkers"*. In *Figure Eight Platform* each $\langle T, H \rangle$ pair is a unit. The *"Turkers"* are asked to select either one of the following three classifications:

- *Textual Entailment by Generality* (TEG),
- *Textual Entailment, without Generality* (TEnG),
- *Other*,

whichever is most appropriate for the $\langle T, H \rangle$ pair under consideration.

The work involved in the annotation of the entailment cases of **RTE-1** through **RTE-5** datasets with TEG, TEnG and *Other* labels. As summaries of Table 2, of the total of 2000 $\langle T, H \rangle$ pairs known to be in entailment relation were uploaded, from which 1740 were submitted for evaluation, having 260 *Gold* pairs contained.

**Table 2.** Summary of TE Recognition (RTE) by generality corpus annotation task.

| | |
|---:|:---|
| **# Input Pairs** | 2000 (RTE-1: 400 + RTE-2: 400 + RTE-3: 400 + RTE-4: 500 + RTE-5: 300) |
| **# Pairs to Launch** | 1740 |
| **# Gold Pairs** | 260 |
| **# Output Pairs** | 1203 |
| **# Discarded Pairs** | 797 |
| **Evaluation Time** | ≈43 days |
| **# *Trusted "Turkers"*** | 2308 |
| **# Trusted Judgments** | 5220 (1740*3) |
| **# Untrusted Judgments** | 60,482 |

We can see that 1203 $\langle T, H \rangle$ pairs were annotated as being TEG. Each pair was evaluated by three *Turkers* and the final average inter-annotator agreement of 0.8 was verified.

This task proved to be hard for the *Turkers*, as it is difficult for a human annotator to identify entailment relation and entailment by generality in particular. This is evidenced by the time spent to complete the task (*Evaluation Time*) and the total number of *Judgments* (*Trusted* + *Untrusted*) needed to achieve the final objective.

Thus, this manually annotated corpus is the first large-scale dataset containing a reasonable number of TEG pairs, representing one of the contributions of our work. This is a valuable resource available to the research community.

## 6. Experimentation

### 6.1. Levels of Word Granularity

In our work, we experimented with two approaches for selecting the words for the calculation of the asymmetry between sentences, to detect relations of textual generality. Thus we can assess which approach has the best performance to identify entailment by generality. In the first approach, we chose to do the calculations without restrictions, i.e., do the calculations with all the words.

In the second approach, we introduce the concept of Multiword Units (MWU), identified the MWU in sentences for the calculation of the asymmetry. Purely linguistic systems follow the first part of the definition of MWU proposed in [43]: an MWU *is defined as a sequence of two or more consecutive words, that has characteristics of a syntactic and semantic unit, and in which the exact and unambiguous meaning or connotation cannot be directly derived from the meaning or connotation of its components.* By definition, MWU is words that co-occur together more often than they would by chance in a given domain and usually convey conceptual information [44]. For example, the French expression *tomber dans les pommes (too faint)* is a sequence of words of which the meaning is non-compositional, i.e., it can not be reproduced by the sum of the meanings of its constituents and thus represents a typical MWU. MWU include a large range of linguistic phenomena as stated in [45], such as compound nouns (e.g., *chantier naval meaning* in French *shipyard*), phrasal verbs (e.g., *entrar em vigor meaning* in Portuguese *to come into force*), adverbial locutions (e.g., *sans cesse meaning* in French *constantly*), compound determinants (e.g., *un tas de meaning* in French *an amount of*), prepositional locutions (e.g., *au lieu de meaning* in French *instead of*), adjectival locutions (e.g., *a longo prazo meaning* in Portuguese *long-term*) and institutionalized phrases (e.g., *con carne*). In our work, for extraction MWU in first five RTE dataset test, used the Software for the Extraction of N-ary Textual Associations (SENTA) [46], which is parameter-free and language-independent thus allowing the extraction of MWU from any raw text. SENTA shows many advantages compared to different methodologies presented so far. It is parameter-free, thus avoiding threshold tuning. It can extract relevant sequences of characters, thus allowing its application to character-based languages.

In summary, our experiments are based on two approaches to the calculations: using all words, and using Multiword Units.

Sample of Calculation for Identify Entailment by Generality

In this section we present our methodology to identify entailment by generality between two sentences, we now apply our methodology on a pair of $\langle T, H \rangle$ extracted from the RTE-3 test set:

*"<pair id="217" entailment="YES" task="IR" length="short" >*
*<t>Pierre Beregovoy, apparently left no message when he shot himself with a borrowed gun.</t>*
*<h>Pierre Beregovoy commits suicide.</h>*
*< /pair>"*

The AAM we use in this demonstration is the Conditional Probability (Equation (2)), for the calculations of the three approaches, the terms and their web frequencies. For the calculations, we used the *HULTIG-C API* (`http://hultigcorpus.di.ubi.pt/`) to calculate all joint and marginal frequencies, so, instead of relying on a closed corpus and exact frequencies, we based our analysis on the Web hits, i.e., the total number of documents where words appear. The following figures illustrate the existing links between sentence terms for both approaches.

The next Figure 1 illustrates the links between sentences when the calculations are made with all terms—*All Words* that compose theses sentences.

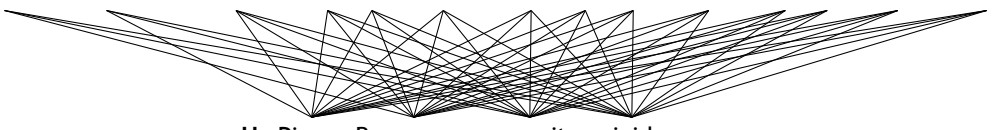

**Figure 1.** Complete unigram connections between two sentences.

In this case, $AISs(T\|H) = 0.26$ and $AISs(H\|T) = 0.20$, then $AISs(T\|H) = 0.26 > AISs(H\|T) = 0.20$, thereby allowing us to conclude that $T$ does not entail $H$.

Through the concept of MWU, it gives us the possibility of the links are not only in single words but also in terms, as we see below (Figure 2), the example has two MWU—*"Pierre Beregovoy"* and *"with a"*.

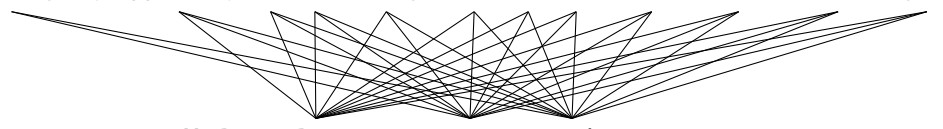

**Figure 2.** Multiword units connections between two sentences.

In this case, $AISs(T\|H) = 0.17$ and $AISs(H\|T) = 0.22$, then $AISs(T\|H) = 0.17 < AISs(H\|T) = 0.22$, so we conclude that $T$ entails $H$ by generality ($T \overset{G}{\to} H$).

## 7. Evaluating the Performance

With the evaluation, the performance of our methodology will help us define what approach—with all words; with MWU—and what AAM—the *Added Value* (Equation (3)), the *Braun-Blanket* (Equation (4)), the *Certainty Factor* (Equation (5)), the *Conviction* (Equation (6)), the *Gini Index* (Equation (7)), the *J-measure* (Equation (8)), the *Laplace* (Equation (9)), and the *Conditional Probability* (Equation (2))—would lead to better recognition of textual entailment by generality.

With this new definition, we know how to implement the future framework and/or toolkits unsupervised and language-independent, with different objectives in NLP.

Our evaluation is based on the analysis of the results obtained through the measures that we present below. The calculation of these measures is based on the Confusion Matrix (CM).

### 7.1. Measures to Evaluate the Performance

Classification or categorization is the task of assigning objects from a universe to two or more classes or categories. In the field of Artificial Intelligence, a CM is a visualization tool typically used in supervised and unsupervised learning. Each column of the matrix represents the instances in a predicted class, while each row represents the instances in an actual class. One benefit of a CM is that it is easy to see if the system is confusing two classes (i.e., commonly mislabeling one as another).

When the dataset is unbalanced (when the number of samples in different classes vary greatly), the error rate of a classifier is not representative of the actual performance of the classifier.

The entries in the confusion matrix have the following meaning in the context of our study: (i) TP is the number of correct **predictions** that an instance is an **Entailment**, (ii) FP is the number of incorrect **predictions** that an instance is **No Entailment**, (iii) FN is the number of incorrect **predictions** that an instance in **Entailment** and (iv) TN is the number of correct **predictions** that an instance is **No Entailment**.

For binary classification, classifiers are typically evaluated using a table of counts like Table 3. An important measure is classification *Accuracy* (AC) and *Precision* (P), which is defined in Equations (16) and (17), respectively. The AC is the proportion of the total number of correct predictions. It is determined using Equation (16). P is defined as a measure of the proportion of selected items that the system got right Equation (17).

$$AC = \frac{TP + TN}{TP + FP + FN + TN} \tag{16}$$

$$P = \frac{TP}{TP + FP} \tag{17}$$

**Table 3.** Contingency table for evaluating a binary classifier. For example, a is the number of objects in the category of interest that were correctly assigned to the category [47].

|  | YES Is Correct | NO Is Correct |
|---|---|---|
| **YES was assigned** | TP | FP |
| **NO was assigned** | FN | TN |

To evaluate the overall performance of our experiments, we used two types of averaging of the previous measures: by calculating the arithmetic average and also through the calculation of the weighted average. In the weighted average, there is some variation in the relative contribution of individual data values to the average. Each data value ($X_i$) has a weight assigned to it ($W_i$). Data values with larger weights contribute more to the weighted average and data values with smaller weights contribute less to the weighted average. The weighted average formula is used to calculate the average value of a particular set of numbers with different levels of relevance. The relevance of each number is called its weight. The weights should be represented as a percentage of the total relevancy. Therefore, all weights should be equal to 100%, or 1. The most common formula used to determine an average is the arithmetic mean formula. This formula adds all of the numbers and divides by the number of numbers. For example, the average of 1, 2 and 3 would be the sum $1 + 2 + 3$ divided by 3, which would return 2. However, the weighted average formula looks at how relevant each number is. Say that 1 only happens 10% of the time while 2 and 3 each happen 45% of the time. In this case, the weighted average would be 2.35. More specifically, in our work, we have defined and used the following equations—*Average Accuracy* (Equation (18)), *Average Precision* (Equation (19)) and *Weighted Average Accuracy* (Equation (20)), *Weighted Average Precision* (Equation (21)).

$$\overline{AC} = \frac{\sum_{i=1}^{n} AC_i}{n} \tag{18}$$

$$\overline{P} = \frac{\sum_{i=1}^{n} P_i}{n} \tag{19}$$

$$\overline{AC}_w = \frac{\sum_{i=1}^{n} AC_i W_i}{\sum_{i=1}^{n} W_i} \tag{20}$$

$$\overline{P}_w = \frac{\sum_{i=1}^{n} P_i W_i}{\sum_{i=1}^{n} W_i} \tag{21}$$

*7.2. Results in TEG Corpus*

In order to assess the effectiveness and general quality of our proposed measures for TEG identification, we have tested our proposed *AISs* measure with all word-similarity functions.

The evaluation functions used are based on the confusion matrix, in particular, the accuracy and the precision. More specifically, we dealt with *Average Accuracy*, *Average Precision*, *Weighted Average Accuracy* and *Weighted Average Precision*.

Here, we report the obtained results of our methodology on the TEG corpus. These are the results we are most interested in, as they concern the problem on which we are focusing our attention, namely identification of entailment by generality.

Concerning **AC**, as seen in Figure 3, the best performance, 0.88, is achieved by the measure *Braun-Blanket* in conjunction with the *MWU* method. The second-best measures are *Added Value* and *Gini Index* with an accuracy of 0.71 . It is important to highlight the significant difference between these two AAM.

In the *"All Words"* approach, we have two measures with the best performance, *Conviction* and *J-measure*, achieving an accuracy of 0.73 and 0.70, respectively. In Figure 3, we realize that although *Conviction* is the best measure with *"All Words"* concerning **Accuracy**, its performance is virtually equivalent to that of a random guesser, for the *"With MWU"* approaches.

In terms of **P**, the *Braun-Blanket* measure, in conjunction with the *MWU* approach, achieved the best results, for both entailment types: *"Entailment by Generality"* (**TEG**) and *"Entailment, without Generality"* (**TEnG**), with, respectively, 0.94 and 0.74 points. For the *All Words* approach, the *Added Value* measure and *J-Measure* achieved 0.84 in **TEG**. To **TEnG**, the best measure is *Conviction* with 0.67 (see Figures 4 and 5).

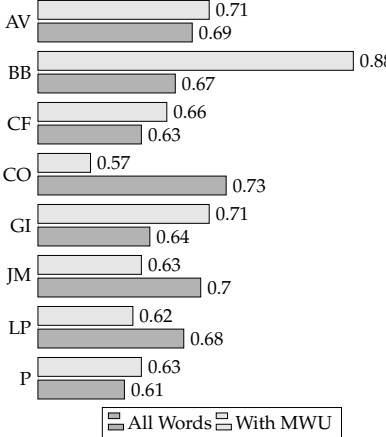

**Figure 3.** Accuracy by Asymmetric Association Measures (AAM).

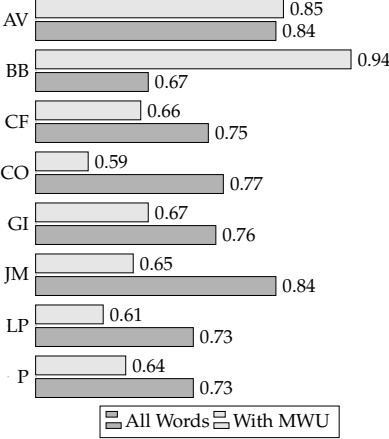

**Figure 4.** Precisions for Textual Entailment by Generality (TEG).

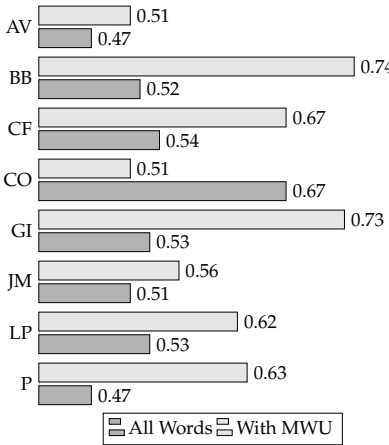

**Figure 5.** Precisions for Textual Entailment, without Generality (TEnG).

With these results for **TEG** Corpus, we conclude that the *Braun-Blanket* measure, in conjunction with the *MWU* approach, presents good results and can detect relations of textual generality in the English language.

*7.3. Results in Portuguese TEG Corpus*

In this section, we present the results of an experiment parallel to the one discussed previously. The main idea was to measure the degree to which our methodology is capable of recognizing TEGs in a different language. For this, we have randomly selected a subset of 1000 $\langle T, H \rangle$ pairs from the TEG Corpus, preserving the proportion of 600 $\langle T, H \rangle$ TEG pairs (Entailment by Generality) and 400 TEnG $\langle T, H \rangle$ pairs (Entailment, without Generality). This subset of 1000 TE pairs was translated into Portuguese.

Concerning **Accuracy**, the best performance is achieved with the measure *Braun-Blanket* in conjunction with the approach **With MWU**, with a result of 0.81, as evidenced in Figure 6. With **"All Words"**, the measure with the best **Accuracy** is *J-measure*, with 0.73.

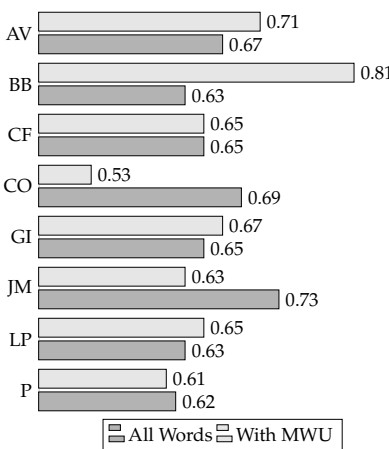

**Figure 6.** Accuracy by AAM.

Concerning **"Precision—Entailment by Generality"** the measure *Braun-Blanket* in conjunction with the *With MWU* approach, presents the best results of 0.89, followed by the measure *J-measure* in conjunction with the approach *All Words*, 0.86. The worst result was achieved in *With MWU* by *Conviction*, 0.57 (Figure 7).

Concerning **"Precision—Entailment, without Generality"** the results are markedly lower. The best result was achieved *With MWU* by *Certainty Factor*, with a value of 0.73. Moreover, the worst results are achieved by *Added Value*, 0.43, in *All Words* (Figure 8).

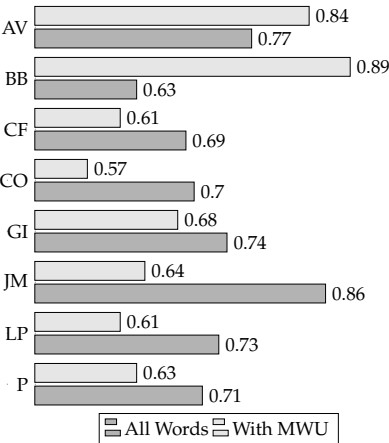

**Figure 7.** Precisions for TEG.

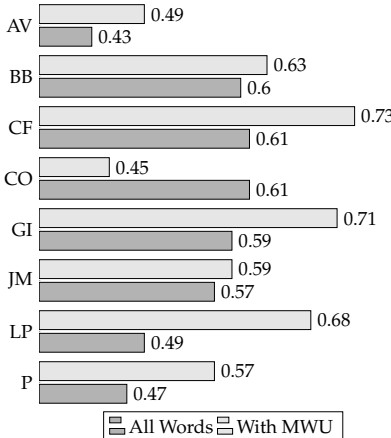

**Figure 8.** Precisions for TEnG.

We are considering the **AC** figures for English and Portuguese, presented in Figure 9, which show a similar scale and variations for the best measure in conjunction with the approach *With MWU*. In terms of precision (**P**), a similar result is achieved. Therefore, we conclude that the performance of our methodology is not significantly influenced by the language, as shown in the following figures.

Both **Accuracy** and **Precision** figures (Figures 9 and 10) show that whether applied to a corpus in English or in the Portuguese language, our methodology provides classification capability significantly better than a random guessing baseline and is virtually indistinguishable concerning the language. This also evidences that our approach is language independent.

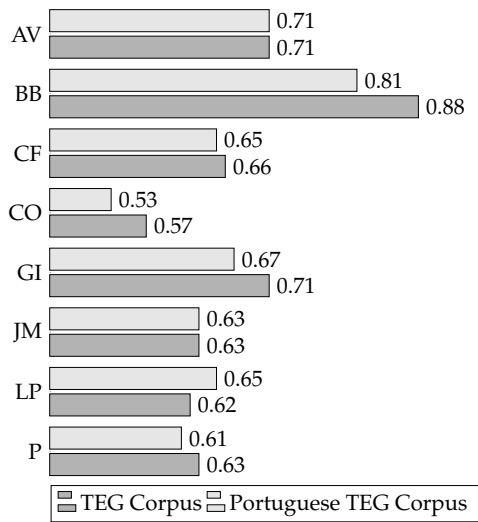

**Figure 9.** Accuracy by AAM, in approach *with Multiword Units (MWU).*

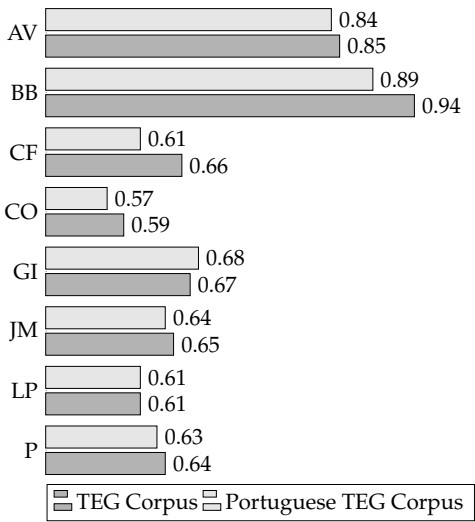

**Figure 10.** Precision for TEG, in approach *With MWU.*

## 8. Conclusions

This work presents a new methodology for recognizing TEG, studies its behavior in a detailed experimental configuration, achieving significant results. As seen in Figures 3–5, there is always a measure and an approach that stands out, namely the *Braun-Blanket* measure **with MWU**. However, *J-measure* and *Conviction* also have good results—(a) *J-measure* in **Precision—Entailment by Generality** with **All Words**, has the second-best performance (with 0.84). In other words, *J-measure* with **All Words** has a good performance to identify entailment by generality between sentences; (b) *Conviction* ranks second for **Accuracy** (with 0.73), and achieves a good result in *Precision—Entailment, without generality or Other*, both by following the approach of using *All Words*.

We can say that our methodology is language independent since comparable results were obtained, relative to those in the English language, although with less significant discrimination between the first and second measures, as shown in Figures 9 and 10. However, in terms of **Accuracy**, Figure 6, and **Precision—Entailment by Generality**, Figure 10, the *Braun-Blanket* achieves the best performance through the approach *With MWU*.

With this paper, we are also contributing to the consideration of a new kind of textual entailment, also providing new experimental resources (TEG Corpus). Our methodology is unsupervised and

language-independent and accounts for the asymmetry of the studied phenomena using asymmetric similarity measures. We have demonstrated through our methodology several exciting results in identifying relations of textual entailment by generality, which indeed are of great importance for many other areas from Natural Language Processing.

**Author Contributions:** Conceptualization, S.P.; methodology, S.P. and G.D.; validation, G.D.; investigation, S.P. and G.D.; data curation, S.P.; writing—original draft preparation, S.P. and G.D.; writing—review and editing, S.P.; supervision, G.D.; funding acquisition, S.P. All authors have read and agreed to the published version of the manuscript.

**Funding:** This research was funded by National Founding from the FCT—Fundação para a Ciência e a Tecnologia, through the MOVES Project—PTDC/EEI-AUT/28918/2017.

**Conflicts of Interest:** The authors declare no conflict of interest.

## Appendix A

### Appendix A.1. RTE Challenges

#### Appendix A.1.1. Evaluation Measures

The evaluation of all runs submitted was automatic, the judgments returned by the system are compared to the Gold Standard compiled by the human assessors. The main evaluation measure was accuracy, i.e., the fraction of correct answers. For the two-way task, a judgment of "NO ENTAILMENT" in a submitted run was considered to match either "CONTRADICTION" or "UNKNOWN" in the Gold Standard.

As a second measure, an Average Precision score was computed for systems that provided as output a confidence-ranked list of all test examples. Average Precision is a common evaluation measure for system rankings and is computed as the average of the system's precision values at all points in the ranked list in which recall increases, that is at all points in the ranked list for which the gold standard annotation is "ENTAILMENT". In other words, this measure evaluates the ability of systems to rank all the $\langle T, H \rangle$ pairs in the test set according to their entailment confidence (in decreasing order from the most certain entailment to the least certain). More formally, it can be written as follows:

$$\frac{1}{R} \sum_{i=1}^{n} \frac{E(i) \times \sharp EntailmentUpTpPair(i)}{i} \tag{A1}$$

where $n$ is the number of pairs in the test set, $R$ is the total number of ENTAILMENT pairs in the Gold Standard, $E(i)$ is 1 if the $i - th$ pair is marked as ENTAILMENT in the Gold Standard and 0 otherwise, and $i$ range over the pairs, ordered by their ranking.

In practice, the more confident the system was that $T$ entailed $H$, the higher the ranking of the pair was. A perfect ranking would have placed all the positive pairs (for which the entailment holds) before all the negative ones, yielding an average precision value of 1. As average precision is relevant only for a binary annotation, in the case of three-way judgment submissions the pairs tagged as CONTRADICTION and UNKNOWN were conflated and re-tagged as "NO ENTAILMENT".

#### Appendix A.1.2. First Challenge

In an overview of the systems participating in the first RTE challenge (http://pascallin.ecs.soton.ac.uk/Challenges/RTE/), of 2005, we saw that the main approaches (the best results) used are based on word overlap [48], statistical lexical relations [24], WordNet [9] similarities [48], syntactic matching [49], world knowledge [24], edit distance between parsing trees [50]. The majority of the systems experiment with different threshold and parameter settings to estimate the best performance. The parameter adjustment process is related to the carrying out of numerous experiments, and still, the settings selected after these experiments may lead to incorrect reasoning.

In [49], the system for semantic evaluation VENSES (Venice Semantic Evaluation System) is organized as a pipeline of two subsystems: the first is a reduced version of *GETARUN*, our system for Text Understanding. The output of the system is a flat list of head-dependent structures with Grammatical Relations and Semantic Roles labels.

The authors of [24] intended to exemplify two different ends of the spectrum of possibilities. The first submission is a traditional system based on linguistic analysis and inference, while the second is inspired by alignment approaches from MT.

In [22], the authors propose a general probabilistic setting that formalizes the notion of TE and describes a model for lexical entailment based on web co-occurrence statistics in a bag of words representation.

The system described in [48] is based on the use of a broad-coverage parser to extract dependency relations and a module that obtains lexical entailment relations from WordNet [9].

The authors of [50] assumed a distance-based framework, where the distance between *T* and *H* is inversely proportional to the entailment relation in the pair, estimated as the sum of the costs of the edit operations (i.e., insertion, deletion, substitution) on the parse tree, which is necessary to transform *T* into *H*. They use different resources to estimate the edit operations cost and to ensure the non-symmetric directionality of the entailment relation.

### Appendix A.1.3. Second Challenge

In the second edition, of 2006, the main directions were generally the same, only algorithms were more sophisticated, and also the results were better (average precision grew from 55.12% in 2005 to 58.62% in 2006). New directions are related to semantic role labeling [51], Machine Learning classification, using background knowledge [52] acquisition of entailment corpora [51]. Some groups tried to detect non-entailment by looking for various kinds of mismatch between the text and the hypothesis.

In [51], they introduce a new system for RTE (known as GROUNDHOG), which utilizes a classification-based approach to combine lexico-semantic information derived from text processing applications with an extensive collection of paraphrases acquired automatically from the WWW, and trained on 200,000 examples of TE extracted from news wire corpora.

The authors of [52] transformed two text snippets into three-layered semantically-rich logic form representations, which generates an abundant set of lexical, syntactic, semantic, and world knowledge axioms and, iteratively, searches for a proof for the entailment between the text *T* and a possibly relaxed version of the hypothesis *H*. They could improve the performance of their system using the lexical inference system in combination with their logical approach.

The system described in [53] defines a cross-pair similarity measure based on the syntactic trees of *T* and *H*, and combines such similarity with traditional intra-pair similarities to define a novel semantic kernel function.

The authors of [54] presented a system of TE based primarily on the concept of lexical overlap. The system begins with a bag of words similarity overlap measure, derived from a combination of WordNet [9] lexical chains to form a mapping of terms in the hypothesis to the source text. It then looks for negations not found in the mapping, and for the lexical edit distance of the mapping. These items are then entered into a decision tree to determine the overall entailment.

The authors of [55] combined two approaches, a shallow method based mainly on word-overlap and a method based on logical inference, using the first-order theorem proving and model building techniques. They used a machine learning technique to combine features from both methods.

### Appendix A.1.4. Third Challenge

In the third edition, of 2007, the groups were oriented on the approaches based on the syntactic structure of the Text (T) and the Hypothesis (H), on semantic understanding of the texts and also on verification of the content and new situations and contexts that meet in the test data.

The GROUNDHOG system [56] uses a pipeline of lightweight, largely statistical systems for commitment extraction, lexical alignment, and entailment classification in order to estimate the likelihood that *T* includes sufficient linguistic content to textually entail *H*.

In [57], the authors include new resources, such as eXtended WordNet [58], which provides a large number of world knowledge axioms, event and temporal information provided by the Temporal Awareness and Reasoning Systems for Question Interpretation (TARSQI) toolkit, logic form representations of events, negation, co-reference and context, and new improvements of lexical chain axiom generation.

The authors of [59] compared $H's$ parse tree against sub-trees of $T's$ parse tree. They transformed the hypothesis making use of extensive semantic knowledge from sources like DIRT [60], WordNet, Wikipedia and a database of acronyms. Additionally, they took advantage of hand-coded complex grammar rules for rephrasing in English.

In [61], two textual entailment approaches are presented. The first one is based primarily on the concepts of lexical overlap, considering a bag-of-words similarity overlap measure to form a mapping of terms in the hypothesis to the source text. The second system is a lexico-semantic matching between the text and the hypothesis that attempts an alignment between chunks in the hypothesis and chunks in the text, and a representation of the text and hypothesis as two dependency graphs. Both approaches employ decision trees as a supervised learning algorithm.

The system presented in [62] has moved from a puristic syntactic approach, in the sense that they only performed dependency parser, to the development of specialized RTE-modules capable of tackling more entailment phenomena. They present a novel approach to RTE that exploits a structure-oriented sentence representation followed by a similarity function.

Appendix A.1.5. Fourth Challenge

The authors of [63] focused on collecting more in-depth semantic features, in using a pipeline of lightweight, largely statistical systems for commitment extraction, lexical alignment, and entailment classification in order to estimate the likelihood that a *T* includes the linguistic content sufficient to entail an *H* textually.

The main idea in [64] is to map every word from the hypothesis to one or more words from the text. For that, this system transforms the hypothesis making use of extensive semantic knowledge from sources like DIRT, WordNet [9], VerbOcean, Wikipedia and a database of acronyms.

The approach proposed in [65] is based on constructing structural features from the abstract tree descriptions, which are automatically extracted from syntactic dependency trees of *T* and *H*. These features are then applied by a subsequence-kernel-based classifier that learns to decide whether the entailment relation holds between two texts.

In [66], they design different strategies to recognize true entailment and false entailment. The similarity between hypothesis and text is measured to recognize true entailment. They detect the same entity and relation mismatch to recognize the false entailment.

The RTE system presented in [67] tackles the entailment phenomenon from two different points of view. First, they build the system's core using several lexical measures and further on, they add some semantic constraints that they think are appropriated for the entailment recognition. The reason for creating this core was given by (i) the fact that the integration of more complex semantic knowledge is a delicate task and it would be easier if they had a stable base system; and (ii) although the proposed core needs some language-dependent tools (e.g., lemmatizer, stemmer), it could be easily ported to other languages.

Appendix A.1.6. Fifth Challenge

In this challenge, the best result for two way is [68], the main idea of their system is to map every word in the hypothesis to one or more words in the text. For that, they transform the hypothesis, using extensive semantic knowledge from sources (as in the previous RTE edition).

The authors of [69] propose a joint syntactic-semantic representation to capture better the key information shared by the pair, and also apply a co-reference resolver to group cross-sentential mentioning of the same entities.

In [70], they propose a new method, SEGraph (Semantic Elements based Graph). This method divides the Hypothesis and the Text into two types of semantic elements: Entity Semantic Element and Relation Semantic Element. The SEGraph is then constructed, with Entity Elements as nodes, and Relation Elements as edges for both Text and Hypothesis. They recognize the textual entailment based on the SEGraph of Text and SEGraph of Hypothesis.

In [71], the authors made use of semantic knowledge-based on Wikipedia. More specifically, they used it to enrich the similarity measure between pairs of text and hypothesis (i.e., the tree kernel for text and hypothesis pairs), with a lexical similarity (i.e., the similarity between the leaves of the trees.

The authors of [72] present an approach to textual entailment recognition, in which inference is based on a shallow semantic representation of relations (predicates and their arguments) in the text and hypothesis of the entailment pair, and in which specialized knowledge is encapsulated in modular components with straightforward interfaces.

**Table A1.** Best RTE systems.

| RTE-1 | | RTE-2 | | RTE-3 | | RTE-4 | | RTE-5 | |
|---|---|---|---|---|---|---|---|---|---|
| Participants | Accuracy | Participants | Accuracy | Participants | Accuracy | Participants | Accuracy | Participants | Accuracy |
| [49] | 0.606 | [51] | 0.754 | [56] | 0.800 | [63] | 0.746 | [68] | 0.735 |
| [24] | 0.586 | [52] | 0.738 | [57] | 0.723 | [64] | 0.721 | [69] | 0.685 |
| [22] | 0.586 | [53] | 0.639 | [59] | 0.691 | [65] | 0.706 | [70] | 0.670 |
| [48] | 0.566 | [54] | 0.626 | [61] | 0.670 | [66] | 0.659 | [71] | 0.662 |
| [50] | 0.559 | [55] | 0.616 | [62] | 0.669 | [67] | 0.608 | [72] | 0.643 |

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
