# Peer review of "Asymmetric Attributional Word Similarity Measures to Detect the Relations of Textual Generality"

_computers, doi:10.3390/computers9040081_

Round 1

Reviewer 1 Report

The paper: “Asymmetric Attributional Word Similarities Measures to Detect Relations of Textual Generality” aims to solve a variant of textual entailment centered around generality of the sentences. The authors propose AISs, an approach for resolution of the considered entailment tasks, alongside with a new data set.

The paper is overall well written, however, the following issues should be addressed prior to potential acceptance:

 L.49: You mention that “It is accepted that textual entailment is not an exact science”. I’d argue that at least to some extent tasks of this type have found their way into mainstream benchmarks, see for example: https://nlp.stanford.edu/projects/snli/. I would suggest that the authors reconsider this claim and instead identify the examples where entailment-based tasks were e.g., presented as shared ones. E.g., t2 in https://sites.google.com/view/mediqa2019 etc.

The related work section is too long, and impacts the reading quality negatively after the incremental re-statements of the five challenges. I would suggest you compactify this section and move parts of it perhaps as the appendix material. Further, all tables could be included in a single float environment, in one row, as they are relatively small.

Sections 3 and 4, albeit not being “Related work”, definitely read like a one. Consider keeping only the key ideas in these sections, and moving the remainder into the appendix -- the reader might have trouble understanding what is a contribution and what is not.

If I understand correctly, the 4.2 is the actual contribution of this work? Could you explicitly state that?

L.473, the sums go up to p, which is the length of the vector X_i. So, the length of X_j is not accounted for? Is there at least an assumption that |X_i| >= |X_j| then?

Define what exactly are X_i and X_j. Based on e.q., I’d assume these are some form of document vectors? This needs to be clarified.

Further, from sec. 6., it looks like AISs is not a vector, but a scalar. Should this be the case, e.q. 12 and 13 are off; assuming X_i and X_j represent e.g., BoW vectors. Consider e.q., 12. Here, the nominator is a scalar. However, the denominator, technically speaking, represents a column vector (+,-,0 relations), which is problematic. Is it possible you in fact meant to write this as a single expression? If this is the case, please, define the denominator (without parenthesis) as some variable, and let AIS = nominator/variable. Much confusion shall be avoided this way. Why must the W_ik \neq W_jl hold? Wouldn’t in most cases  AS(x||x) = 1 anyways? Please, clarify.

The empirical section is interesting, however:

  1. The legends in fig.3 are indistinguishable from one another, so I am not sure what I am looking at.
  2. You experimented on protugese and english corpora, which led you to the claim that this works across languages. Isn’t this a bit too general? Does this work for arabic, chinese and .e.g, swahili too? I am not entirely sure it would, please elaborate.
  3. Finally, I am missing at least one more baseline approach. To my current understanding, it shouldn’t be that much of  a hurdle to also test how e.g., BERT behaves (adapt it from SNLI for example). This would make this section much stronger.

A note on related work:

Even though the paper includes almost 80 citations, key parts of related work, done in the recent work are in my opinion missing, and should be added. For example:

  1. The neural language models were shown to perform well also for entailment-like tasks. Please consult e.g., https://openreview.net/pdf?id=SkZxCk-0Z, https://www.aclweb.org/anthology/C16-1212.pdf, https://arxiv.org/abs/1907.11932, https://www.aclweb.org/anthology/D19-1630/ etc.
  2. Your task closely corresponds to hypernymy relation and its exploitation. See for example https://www.sciencedirect.com/science/article/pii/S0885230820300371, https://bmcbioinformatics.biomedcentral.com/articles/10.1186/s12859-018-2205-3 regarding other potential implications of being able to exploit hypernymy

Overall, should the raised issues be addressed, the paper could be accepted.

Author Response

We are very grateful to the reviewer for his time and comments. We apologize for just now submitting a new version. We re-reflected and answered the suggestions and changed the paper accordingly.

Reviewer 2 Report

This paper discusses the use of asymmetric word similarity measures to detect relations of textual generality (textual entailment by generality).

The topic is relevant and the paper is well written and organized.

The motivation is adequate and the obtained results seem to be interesting.

However, there are several issues that can be improved:

  • It is not clear which are the original contributions of this paper. This should be made clearer in the introduction.
  • Section 2.2. -- There are no references to the more recent work using neural networks and deep learning approaches. I understand this approach is not neural network based but, at least, some references should be done.
  • Section 3 , last paragraph -- It is important to point out that this is not a theorem and it was not proved. This is the assumption the authors use in their work. It can not be said "As a consequence, an entailment ... will hold if and only oif ..."
  • Section 4 -- It is not clear which is the contribution of the authors in this paper. Is it the measure AISs? This should be clarified.
  • Section 5 -- The created corpus (TEG) is an important contribution of this work. This section should be extended with more details and examples.
  • Section 7 -- Accuracy, precision, recall and f-measure are standard measures and micro or macro versions of precision and recall can be used. I suggest to revise this section in order to present these standard and relevant measures.
  • Finally, I strongly suggest the authors to compare the obtained results with other approaches, allowing to better evaluate its performance. For instance, it would be quite simple to create a simple binary classifier (for instance, using SVMs and BERT embeddings to represent the sentences) and compare the obtained results with this approach.

Author Response

(The authors gave the same response as above.)

Round 2

Reviewer 1 Report

The authors have addressed the raised concerns. Prior to publication, I would suggest:

1.) a re-read of the grammar.

2.) The table 2 could be restructured -> only one column is filled with values now. Please consider reporting just this column if that is the case.

3.) Re-check the reference consistency.

Author Response

We are very grateful to the reviewer for his time and comments. We apologize for just now submitting a new version.

Reviewer 2 Report

I believe the authors have successfully answered my comments.

Author Response

We are very grateful to the reviewer for his time and comments.